# The Combination of Amoxicillin and 1,8-Cineole Improves the Bioavailability and the Therapeutic Effect of Amoxicillin in a Rabbit Model

**DOI:** 10.3390/antibiotics11101294

**Published:** 2022-09-22

**Authors:** Ahmed Amin Akhmouch, Soukayna Hriouech, Hanane Chefchaou, Mariam Tanghort, Aouatef Mzabi, Najat Chami, Adnane Remmal

**Affiliations:** 1Department of Biology, Faculty of Science Dhar El Mahraz, University Sidi Mohammed Ben Abdellah, Fez 30050, Morocco; 2Systems and Sustainable Environment Laboratory, Private University of Fez UPF, Fez 30000, Morocco

**Keywords:** 1,8-cineole, amoxicillin, clavulanic acid, pharmacokinetic parameters, bioavailability, combination therapy, *Escherichia coli* ESBL, rabbit model

## Abstract

In this study, the effectiveness of the combination therapy of 1,8-cineole with amoxicillin (AMX) and clavulanic acid (Clav) was investigated. For this, the pharmacokinetic behaviors of AMX in rabbits were studied after a single oral dose. The animals were divided randomly into two groups: the reference group (received AMX/Clav (50/12.5 mg/kg)) and the test group (received AMX/Clav/1,8-cineole (50/12.5/10 mg/kg)). Blood samples were collected prior to administration and after T1h, T2h, T3h, and T6h post-administration. Plasma concentrations of AMX were quantified using a validated HPLC method. The antibacterial activity of plasma and cerebrospinal fluid (CSF) of treated rabbits was tested against *Escherichia coli* ESBL-producing a strain by microdilution method. The obtained results showed significant differences in pharmacokinetic parameters between the two groups. The resulting AUC_0–6h_ and C_max_ mean values of the AMX reference group were 14.74 µg.h/mL and 3.49 µg/mL, respectively. However, those of the AMX test group were 22.30 µg.h/mL and 5.79 µg/mL, respectively. The results showed that the antibacterial activity of the plasma and CSF test group was significantly higher than that of the reference group. The effectiveness of this combination (Olipen: AMX/Clav/1,8-cineole) was demonstrated by increasing the level of the antibiotic and by improving the bioavailability.

## 1. Introduction

For many years now, the positive effects of antibiotics in the treatment of various infectious diseases have been well documented [1]. Antibacterial therapy was totally revolutionized with the discovery of penicillin in 1929 by Fleming [2]. Unfortunately, overuse, inappropriate use, and uncontrolled use are major factors that are responsible for the emergence and spread of antibiotic-resistant strains [3,4]. Recently, about 700,000 deaths have been recorded annually due to antibiotic resistance [5]. According to predictions by the World Health Organization (WHO), the death rate caused by diverse resistant bacteria could reach 10 million per year by 2050 [6,7]. Bacterial meningitis (BM), caused by *Streptococcus pneumoniae*, *Neisseria meningitidis*, or *Escherichia coli*, has high mortality, morbidity, and serious neurological damage sequelae [8]. Clearly, BM was responsible for more than 276,000 deaths around the world [9]. In Morocco, per 100,000 inhabitants the cumulative incidence of bacterial meningitis was 2.9 [10]. To limit the emergence of these bacterial diseases, AMX alone or combined with the β-lactamase inhibitor clavulanic acid (Clav) are the most frequently prescribed antimicrobials in human medicine [11,12]. Extended-spectrum β-lactamases (ESBLs) are enzymes that confer resistance to most β-lactam antibiotics, such as penicillins, including AMX [13]. AMX and/or third generation cephalosporin were the antibiotics of choice for treating all patients with BM [14]. Nevertheless, resistance to AMX/Clav is increasingly being reported [10,12,15]. Thus, the search for an alternative strategy with greater efficacy is an urgent medical need. Several studies have investigated the use of antibiotics in combination with major essential oil (EO) compounds [16,17,18,19,20,21]. They have revealed that this approach could be efficiently used in therapy to overcome any resistance to antibiotics. Nowadays, some researchers reported that essential oils have the capacity to act as resistance modifying agents and thereby enhance the efficacy of antibiotics [20]. Based on these facts and on our previous studies [16,21] of the combination of AMX with 1,8-cineole EO, a major compound used to combat ESBL bacterial resistance, we have conducted an in vivo experiment to study the pharmacokinetics and pharmacodynamics of AMX combined with 1,8-cineole using a rabbit model.

## 2. Results

### 2.1. Chemical Analysis of the Blood Samples

After oral administration of AMX at a simple dose of 50 mg/Kg BW, all rabbits were clinically healthy, and no adverse reactions were observed during the experimental period.

The HPLC method used showed good performance for quantifying AMX, as reported by Pires et al. The mean absolute recovery of AMX in plasma was 93% at 1 μg/mL, 94% at 10 μg/mL, and 95% at 50 μg/mL. The LOD and the LOQ for AMX were 0.1 and 1 μg/mL, respectively. The calibration curve was linear over the range of 1.0 to 50 μg/mL, with a regression coefficient ≥0.999 and an intercept not significantly different from zero.

Figure 1 shows that under the described chromatographic conditions, the retention times for AMX and internal standard were 4.3 and 5.3 min, respectively. As also shown in Figure 1, no endogenous interfering peaks appeared in the retention times of the compounds of interest. The semilogarithmic plot is given in Figure 2.

The animals tolerated the oral application of the two treatments well. The mean pharmacokinetic parameters are given in Table 1. The obtained results of the chemical analysis showed significant differences in pharmacokinetic parameters between the reference and the test groups (*p* < 0.05). The resulting mean values of AMX absorption parameters (AUC_0–6h_, C_max_ and, T_max_) showed that AMX contained in the test treatment was absorbed at a significantly higher rate than AMX of the reference treatment (*p* < 0.05).

Compared to the AMX reference treatment, the amount of AMX test treatment that reached the bloodstream was one and a half times higher. Furthermore, the time taken to reach the C_max_ (T_max_) decreased by about 20%. Moreover, the test group had an extended mean half-life (T_1/2_) of AMX (2.94 h) compared to the AMX reference treatment (2.21 h).

The data in Figure 2 show the mean plasma concentrations of AMX in the reference and test groups. After oral administration, plasma concentrations of AMX could be quantified until 6 h (T6h) post-administration in both groups. However, the mean plasma concentration of AMX was significantly higher in the test group, when AMX was combined with 1,8-cineole, after T1h, T2h, and T6h of administration, compared to AMX/Clav used alone (the reference group).

### 2.2. Biological Analysis of the Blood Samples

Results of the obtained inhibition percentages of *Escherichia coli* ESBL-producing strain by both tested treatments, at time zero, T1h, T2h, T3h, and T6h are illustrated in Figure 3. As shown at T0, no significant difference (*p* > 0.05) was observed on the antibacterial activities of plasma samples between the two groups. At T1h and T6h after the treatment of animals with AMX/Clav or AMX/Clav/1,8-cineole, the observed antibacterial activity of the plasma of the test group was significantly higher (*p* < 0.05) than that of the plasma of the reference group. In contrast, the growth inhibition of the test group at T2h and T3h was slightly higher than that of the reference group, but no significant statistical difference was observed. The most effective result was observed for the plasma of the test group at T6h.

### 2.3. CSF Samples Biological Analysis

Anti-*E. coli* activity of the cerebrospinal fluid obtained from treated rabbits at different post-administration time points (T2h, T4h, and T6h) is shown in Figure 4. Two hours post-administration, the inhibition percentage of *E. coli* in the CSF of the test group that received AMX/Clav combined with 1,8-cineole was 77%. This was significantly (*p* < 0.001) higher compared to the reference group, which caused only 31% growth inhibition. The test group showed a peak inhibition of 90% in CSF four hours (T4h) after administration compared to the reference group (44%). At T6h, the bactericidal activity of the CSF in both groups decreased compared to T4h. However, the percentage inhibition in the test group (49%) was still significantly (*p* < 0.05) higher than that of the reference group (14%).

## 3. Discussion

Antibiotics such as β-lactams (AMX), vancomycin, sulfamethoxazole, and cephalosporins are often prescribed to treat bacterial infections—inter alia, meningitis [22]. However, the clinical efficacy of any antibiotic against meningitis depends, in addition to its bioavailability, on its ability to penetrate the blood–brain barrier (BBB) which is the major limitation of the efficient delivery of the therapeutic agents [23]. Thus, the success of an antibiotic in the CSF is influenced by its molecular size, lipophilicity, plasma protein binding, and affinity to transport systems at the BBB [24]. Moreover, even if the antibiotic has the capacity to cross the BBB and reach the CSF, it may prove ineffective not only because of bacterial resistance but also because the concentration that reaches the CSF is insufficient. All these reasons led to changing the therapeutic protocols, and there was recourse to increase the recommended dose, implement bi-therapy of antibiotics, and sometimes use tri-therapy to overcome this problem of bacterial meningitis (BM). A dose of β-lactam or other antibiotics higher than what is commonly recommended can result in much higher CSF levels [22]. In a pharmacokinetic analysis, 5 g of β-lactam meropenem every 8 h was required to achieve CSF trough concentrations (>2 mg/L) in 95.1% of patients [25]. Consequently, even the 2 g of meropenem taken every 8 h was not likely to achieve a reasonable target concentration in the CSF, particularly in infections with Gram-negative strains [26]. In the case of multi-resistance, the intrathecal administration of antibiotics must be acomplished by the intravenous route [22]. Costerus et al. [14] chose to administer AMX and/or cephalosporin (3G) to all patients with BM. MacDougal et al. [27] combined the administration of ampicillin, vancomycin, and cephalosporin (3G) as recommended empirical therapy. Various combinations of AMX to Clav have been introduced worldwide to enhance the practicality of dosing and of improving treatment recommendations for more severe infections or those caused by resistant bacteria [28,29]. Therefore, finding an alternative, urgently needed, and effective drug to overcome the diverse bacterial infections seems to be crucial. The aim of the present research paper was designed to verify if the synergistic effect of the AMX/Clav and 1,8-cineole combination obtained in vitro [21] would be reflected in vivo in the rabbit model. A rabbit was more adequate for the performance of the present experiments than mice and rats, because the size of the rabbit allowed us to recover sufficient quantities of physiological fluids to test the antimicrobial activity of the drugs using a microtechnique. Results from our previous in vitro findings demonstrated that the addition of 1,8-cineole significantly increased the antibacterial activity of AMX [21]. Previous research data revealed that 1,8-cineole was a strong active compound against many pathogenic strains, including *Escherichia coli*, *Staphylococcus aureus, Pseudomonas aeruginosa*, and *Bacillus subtilis* [30,31]. Moreover, 1,8-cineole had strong effect against *Bacillus cereus* and *Cryptococcus neoformans* at a dose of 2 mg/mL [32]. A trial conducted by Moo et al. [33], illustrated that 1,8-cineol has potent activity against carbapenemase-producing *Klebsiella pneumoniae* (KPC-KP) at 28.83 mg/mL. In this study, the authors demonstrated that 1,8-cineol induced oxidative stress and membrane damage, resulting in KPC-KP cell death. This was observed by scanning and transmission electron microscopies, which confirmed bacterial cell membrane rupture and loss of intracellular materials [33]. It is well known that there are limits to the use of essential oils and their main compounds because of their toxicity. For 1,8-cineole, as reported previously, the LD50 value for oral administration of 1,8-cineole is 1280 mg/kg in rats [34]. Oral administration of 1,8-cineole at 100 mg/kg did not cause any visible toxic symptoms, respiratory distress, ataxia, convulsion, or mortality in the animals [34]. The LD50 values have been reported to be 2300 mg/kg (intramuscularly) for the Guinea pig, 1500 mg/kg (subcutaneously) for dogs, and 50–3500 mg/kg for mice [35]. Thus, the concentration used in the present work (10 mg/kg BW) was considerably lower than the LD50 used in the previous studies.

Over the years, researchers have shown great interest in the use of 1,8-cineole with chemical antimicrobial agents such as chlorhexidine digluconate [36] and mupirocin [37]. This aims to enhance their antibacterial activities, in the treatment of vulvovaginal candidiasis, bacterial vaginosis, and acne [38]. Findings in vitro by Kwiatkowski et al. [39] and those in vitro and in vivo by Hriouech et al. [40] revealed the synergistic effect of 1,8-cineole in combination with beta-lactam antibiotics, including penicillin G and AMX, against MRSA strains.

In the current research work, the pharmacokinetic profile of AMX in rabbits was evaluated in both groups, after administration of a single dose of 50 mg/kg BW. The determination of AMX plasmatic concentration was based on an HPLC method. Partial analytical validation was performed to adapt the designed HPLC method to AMX quantification in rabbit plasma. Unfortunately, the mean CSF concentration of AMX in the reference and the test groups could not be measured by HPLC due to technical reasons.

Data demonstrated a significant increase in AMX plasma concentration in the rabbits. The 1,8-cineole increased the rate and extent of AMX absorption, increasing the C_max_ from 3.49 ± 0.2 to 5.79 ± 0.2 µg/mL, and AUC_0–6h_ increased from 14.74 ± 0.9 to 22.30 ± 0.4 µg/mL·h. In the pharmacokinetic swine model created by Sun et al. [41], the mean maximal plasma concentration of AMX reached 2.58 μg/mL after 1.14 h, following AMX oral administration at a dose of 10 mg/kg. A study performed by Ali et al. [42] demonstrated that the extract of *Nigella sativa* significantly increased the permeation of AMX, using an in vitro everted rat intestinal sac model. Additionally, the same authors showed that this combination increased the rate and extent of AMX absorption in rats, increasing the C_max_ from 4.138 to 5.995 µg/mL and the AUC_0-t_ from 8.89 to 13.483 µg/mL·h. Likewise, Jiang et al. [43] reported that co-administration of 3% 1,8-cineole and a mixture of antibiotics (0.1 g/L neomycin and 0.1 g/L ampicillin) significantly increased chicken weight and the *Lactobacillus* spp. (*Lactobacillus aviarius*) proportion; decreased *Escherichia* spp. and *Enterococcus* spp. Proportions; and decreased proinflammatory markers (NF-κB, TNF-α, IL-1β, and IL-6) in the upper ileum. On the other hand, in recent research, Kardos et al. [44] suggested that antiviral treatment with additional 1,8-cineole caused a significant reduction in cough frequency and other symptoms of acute bronchitis compared to antiviral treatment alone. The difference between our results and previous reports might be attributed to the dosage, the animal species, and the different formulations. These results showed that the combination of 1,8-cineole with AMX improved the bioavailability of AMX partly by improving intestinal absorption, and mainly by prolonging its half-life. The extension of the half-life through the combination with 1,8-cineole means that the treatment can be optimized by changing the dose and frequency of the drug. This optimization will have a positive impact on reducing the risk of toxicity and side effects.

The half-life of amoxicillin after oral administration has been reported by many studies. In animals, AMX has a mean elimination half-life of approximately 1 h [45,46,47].

In the present research, the antibacterial activities in both plasma and CSF illustrated a higher therapeutic effect of AMX/Clav combined with 1,8-cineole compared to AMX/Clav alone. The plasma AMX concentration was positively correlated with both CSF and plasma bactericidal activity. This increase in biological activity in the test group could be explained by an improvement in the bioavailability of the antibiotic through interactions with 1,8-cineole thereby increasing its intestinal absorption and diffusion across the blood–brain barrier (BBB). This can be also explained by the fact that the association of AMX and 1,8-cineole protects the antibiotic by considerably decreasing ß-lactam’s affinity for ß-lactamase, as we previously demonstrated with the enzymatic assay [21]. Thus, AMX recognition changes in the presence of 1,8-cineole according to these results. It is known that the beta-lactams like penicillin are the substrate of OAT transporters [48]. Since 1,8-cineole decreased ß-lactamase’s affinity for AMX, it could even decrease that of OAT transporters. Our results are in alignment with earlier research, which reported the efficacy of classical β-lactams in combination with β-lactamase inhibitors in treating anthrax meningitis [49]. In addition, Jamshidi et al. [29] reported that the concomitant administration of thymol, an EO major component, at the 50, 150, or 300 mg/Kg of body weight with AMX/Clav can cause a significant enhancement. All data were supported by a spectroscopic assay that demonstrated that the addition of 1,8-cineole allowed the rearrangement of AMX molecules in the form of oligomers of 3–4 amoxicillin molecules [50]. This rearrangement led to the formation of a new complex that penetrated easily and exerted strong antimicrobial action. As the two molecules 1,8-cineole and AMX are slightly hydrophobic, and since their association forms a complex more hydrophobic than either of the two molecules, it can cross the membrane more easily [50].

This new concept was tested on humans (clinical trials). The results obtained (article in process of being written) opened the path to a new drug currently commercialized by Sothema under the name Olipen (AMM: DMP/21/NNPdDMP/VHA/18). This innovative pharmaceutical product is used in therapy to overcome bacterial resistance to ß-lactam antibiotics whilst reducing the rate of resistance.

## 4. Materials and Methods

### 4.1. Drugs and Reagents

A commercial formulation of AMX/Clav oral powder (Soclav 500 mg/62.5 mg; SOTHEMA, Bouskoura, Morocco) was used for the treatment of the animals. Then it was dissolved in sterile distilled water and stirred until totally dispersed. The AMX/Clav was administered at the dose of 50/12 mg/kg BW.

The EO major compound, 1,8-cineole, obtained from Sigma Aldrich (Saint-Quentin-Fallavier, France) was dispersed in a viscous solution of 0.2% (*v/v*) agar according to the method described by Remmal et al. [51]. Then, it was added to AMX/Clav and mixed before administration to animals (AMX/Clav/1,8-cineole at the dose of 50/12/10 mg/kg). These components (AMX/Clav/1,8-cineole) are the active pharmaceutical ingredients (API) of the new clinically approved drug Olipen^®^ (Sothema, Bouskoura, Morocco).

All used reagents were of analytical grade. Acetonitrile (ACN), methanol (MeOH), AMX, and cefadroxil (internal standard) were purchased from Sigma Aldrich (Saint-Quentin-Fallavier, France). Sodium phosphate salt (NaH_2_PO_4_) was purchased from Merck Millipore (Molsheim, France). The deionized water was prepared using a Milli-Q system (Millipore, Molsheim, France).

The stock solution of AMX (1 mg/mL) was prepared in H_2_O:can (95: 5). The working standard solution (100 µg/mL) was prepared from the stock solution by 1/10 dilution in H_can_ACN (95:5). The stock solution of cefadroxil was prepared in MeOH (1 mg/mL). Stock and working standard solutions were aliquoted in propylene tubes, protected from light, and stored at −20 °C until used.

### 4.2. Animals

Twelve 73-day old, female New Zealand rabbits were provided by the Atlas Rabbits Cooperative of Morocco. The animals were randomly divided into two groups of six and placed in cages. All groups were housed in the animal home and were given feed and water ad libitum. The rabbits were clinically healthy and were treated in accordance with the National Health and Research Council Ethics Committee guidelines. The experiments started after a one-week acclimatization period.

### 4.3. Experimental Design

The weight of each animal in each group was measured before treatments started. The weight of each animal was roughly 2 kg. Rabbits of the first group, known as “the reference group,” received a single dose of 50 mg/Kg BW of AMX and 12.5 mg/kg BW of Clav (AMX/Clav). Rabbits of the second group, known as “the test group,” received a single dose of 50 mg/kg BW of AMX and 12.5 mg/kg BW of Clav, supplemented with 10 mg/kg BW of 1,8-cineole (AMX/Clav/1,8-cineole). All treatments were given orally.

### 4.4. Sampling of Blood and Cerebrospinal Fluid (CSF)

Blood samples were collected at T0h, T1h, T2h, T3h, and T6h after treatment administration from the two groups of rabbits. The samples were obtained from vein auricularis taken in tubes already containing the anti-coagulant lithium heparin (2.5 mL, AX3430, LABO-MODERNE, Gennevilliers, France). The collected blood samples were centrifuged at 14,000 rpm for 15 min, and the plasma was collected and stored at −80 °C for further analysis [41].

CSF samples from the two rabbit groups were obtained by intracisternal taps, before and after the administration of treatments T0h and at T2h, T4h, and T6h. Animals were anesthetized by intramuscular injections of ketamine (35 mg/kg) (Sigma Aldrich, France) plus 2.5 mg/kg of xylazine (5 mg/kg) (Sigma Aldrich, Saint-Quentin-Fallavier, France). Furthermore, approximately 0.25 mL of CSF was drawn from each animal. The CSF was transferred to a new tube and was protected from the light and frozen at −80 °C until use [43].

### 4.5. HPLC Analysis

HPLC analysis was used to ascertain the AMX concentrations in the plasma. The applied method is described in detail by Pires et al. [52], and was previously validated according to FDA Guidance for Industry Bioanalytical Method Validation (2018) [53]. AMX and cefadroxil (internal standard) were extracted from plasma samples by protein precipitation. For each rabbit, standard solutions were prepared in Eppendorf microtubes by adding 20 µL of plasma (collected at time zero), 20 µL of internal standard working solution (1 mg/mL), 10 µL of AMX working solution (100 µg/mL), and 50 µL of cold MeOH (Kept on ice). Assay solutions were prepared in Eppendorf microtubes by adding 20 µL of plasma (collected at T1, T2, T3 or T6), 20 µL of internal standard working solution (1 mg/mL), and 60 µL of cold MeOH. After brief vortex mixing of all solutions, the microtubes were centrifuged (14,000 rpm at 4 °C for 15 min). The supernatants were transferred to the injection vials, and 20 µL was injected into the chromatographic system.

The analyses were performed on a WATERS ALLIANCE 2690 chromatographic system equipped with an LC pump, an injector, a thermostat Ed LC oven, and a diode-array detector (Waters, Milford, MA, USA). The drug analysis data were acquired and processed using EMPOWER III software. The mobile phase was a mixture of phosphate buffer (0.01 mol/L), pH = 4.8 and ACN (95:5 *v/v*), pumped in isocratic mode, at a flow rate of 1.3 mL/min through the column (Lichrosorb^®^ 10µm RP 18, 250 mm × 4.6 mm; Phenomenex, Saint-Quentin-Fallavier, France) at room temperature. Peaks were monitored by UV absorbance at 229 nm. The plasma concentration of AMX (µg/mL) was calculated by the following formula:AMX (µg/mL) = (RE × Conc T × FDE ×Act (%))/(RT × 100)(1)
RE: average area ratio of AMX/cefadroxil in the test solution; RT: average area ratio of AMX/cefadroxil in the control solution; Conc T: concentration (µg/mL) of AMX in the control solution; FDE: dilution factor of plasma in test solution; Act%: activity of AMX reference standard (%).

### 4.6. Pharmacokinetic Analysis

Maximum observed plasma concentration (C_max_) and time taken to reach it (T_max_) were obtained from drug concentration vs. time curves. The areas under the AMX concentration vs. time curves from 0 to 6 h (AUC_0–6h_) were calculated using the trapezoidal method, and the first-order elimination rate constant (Ke) was estimated using the least-square regression of the points describing the terminal log-linear decaying phase. T_1/2_ was derived from Ke (T_1/2_ = ln2/Ke).

### 4.7. Blood or CSF Samples and Microbiological Assay

The plasma or CSF antibacterial effect was determined using a microbiological assay method with *Escherichia coli* ESBL-producing strain, obtained from the National Institute of Hygiene (INH, Rabat, Morocco, as a reference organism). The stock culture was kept at −20 °C. The inoculate used was obtained from overnight cultures grown on sterile Mueller–Hinton Broth (MHB) (BIOKAR, Allonne, France) at 37 °C. Using optical density (OD) at 540 nm; the concentration of microorganisms in each overnight suspension was determined using previously established OD/concentration standard curves. The suspensions were then diluted in MHB to obtain a final inoculum size of 2 × 10^7^ organisms per mL (CFU/mL).

The effect of each plasma or CSF sample against the *Escherichia coli* strain growth was detected by the microdilution method [54,55]. All samples were serially diluted with MHB using sterile 96-well microplates (Iwaki microplate, Asahi Techno Glass, Chiba, Japan). For each rabbit of the different treatment groups, ten horizontal wells per rabbit were prepared. Thus, we took 100 µL of MHB, 50 µL of the samples collected at time zero, T1h, T2h, T3h, or T6h for plasma; and at time zero, T2h, T4h, or T6h for CSF. The volume of 50 µL from the suspension of micro-organisms (2 × 10^7^ CFU/mL) was mixed in duplicate. This suspension was diluted in 96-well plates at a density of 5 × 10^6^ CFU/well. A negative control was prepared in duplicate by adding 200 µL of MHB to the twelve horizontal wells. A positive control was prepared in duplicate by adding 150 µL of MHB and 50 µL from the suspension of microorganisms (2 × 10^7^ CFU/mL) to the twelve horizontal wells.

Using a microplate spectrophotometer (Versamax, Molecular Devices. USA), the optical densities (OD) were determined at 540 nm at time zero and 24 h after incubation of the microplates at 37 °C. The inhibition percentage of the *Escherichia coli* in the presence of each rabbit’s plasma or CSF was calculated using the following [54]:% inhibition = [1 − ((OD_T24_ − OD_T0_)/(OD_C24_ − OD_C0_))] × 100(2)
where OD_T0_: OD of experimental well at time zero; OD_T24_: OD of experimental well after 24 h incubation; OD_C0_: OD of positive control well at time zero; OD_C24_: OD of positive control well after 24 h incubation

### 4.8. Ethical Approval

The ethical institutional committee, Faculty of Sciences Dhar El Mahraz, University Sidi Mohamed Ben Abdallah, Fez, Morocco, approved the protocol. All the experimental proceedings performed on laboratory animals followed the internationally accepted standard guidelines for animal care. The authors sought to minimize animal suffering and limited the number of animals used.

### 4.9. Statistical Analysis

All experiments were performed in triplicate, and all results are expressed as average ± standard deviation. The data were analyzed with Student’s tests (Graph Pad Prism, version 5.03). Differences of *p* < 0.05 were considered statistically significant.

## 5. Conclusions

To summarize, our results showed that the pharmacokinetics of AMX/Clav associated with 1,8-cineole administered orally in rabbits demonstrated immediate absorption, slow elimination, and the ability to reach effective treatment concentrations rapidly. Regarding the antibacterial activity of plasma and CSF, the association AMX/Clav/1,8-cineole was greater than that of AMX/Clav alone, indicating that with this combination, AMX could enhance its penetration into tissue and have improved effectiveness.

Based on the results of the current research paper and those obtained previously in vitro, it can be concluded that 1,8-cineole has a potentiating effect on AMX/Clav. Furthermore, the AMX treatment can be moderated by decreasing drug intake frequency and doses.

## 6. Patents

Remmal, A.; Akhmouch, A.A. Pharmaceutical Formulation Comprising Cineole and Amoxicillin. US20190255024. 2019. Available online: https://patentscope.wipo.int/search/en/detail.jsf?docId=US250863563&docAn=16306262 (accessed on 1 June 2022).

## Figures and Tables

**Figure 1 antibiotics-11-01294-f001:**
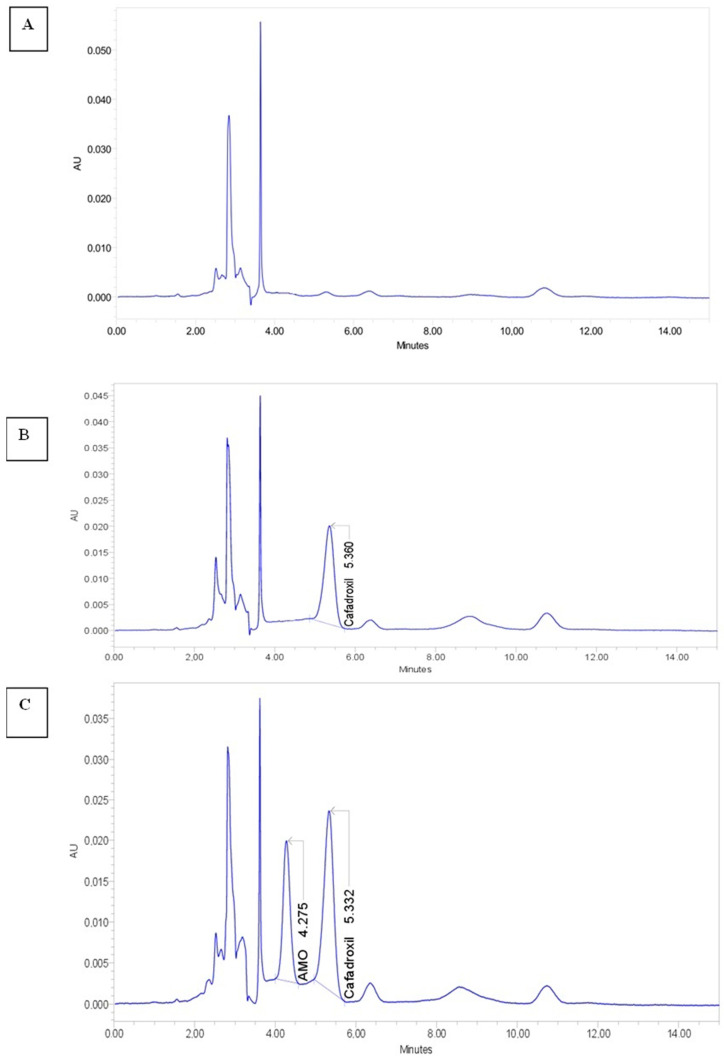
Chromatographic analysis of AMX: (**A**) plasma blank; (**B**) plasma + internal standard (cefadroxil 20 µg/mL); (**C**) plasma + AMX (10 µg/mL) + internal standard (cefadroxil 20 µg/mL).

**Figure 2 antibiotics-11-01294-f002:**
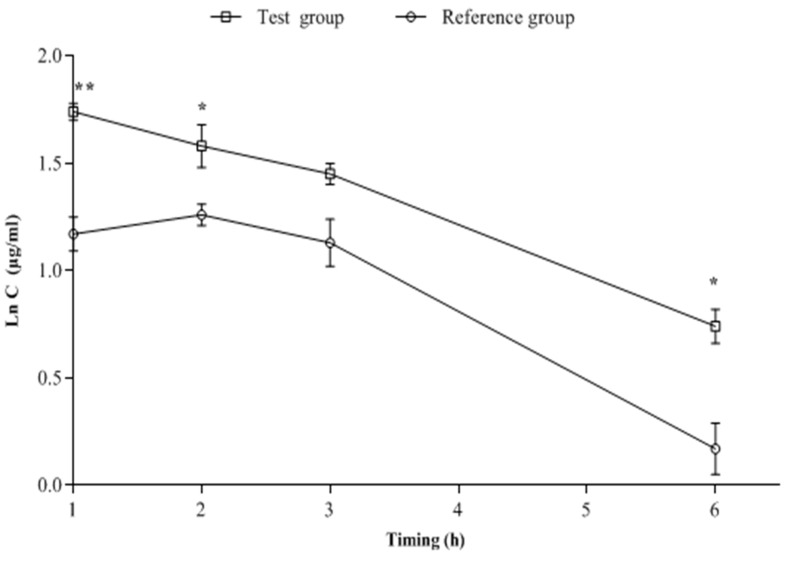
Curves of the mean plasma concentration of AMX for reference and test groups vs. time (h) at a semilogarithmic scale. *: statistically significant difference compared to reference group (*p* < 0.05). **: statistically significant difference compared to reference group (*p* < 0.01).

**Figure 3 antibiotics-11-01294-f003:**
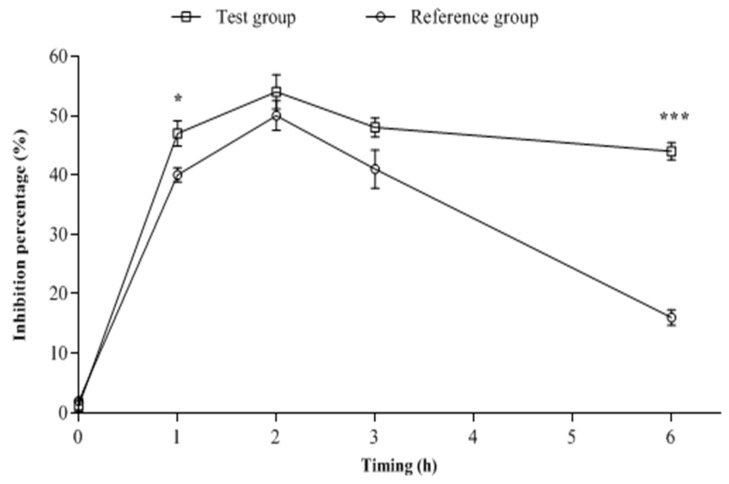
Effects of plasma samples collected from animals treated with AMX/Clav (reference group) or with AMX/Clav/1,8-cineole (test group) against *E. coli* ESBL-producing strain growth (inhibition%). *: statistically significant difference compared to reference group (*p* < 0.05); ***: statistically significant difference compared to reference group (*p* < 0.001).

**Figure 4 antibiotics-11-01294-f004:**
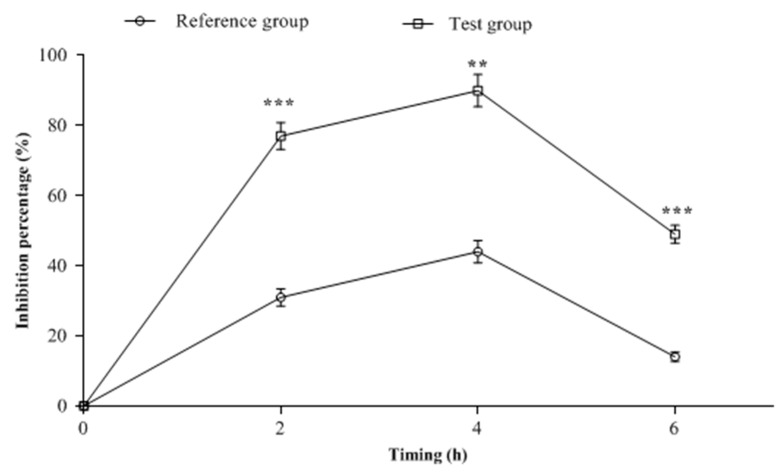
Effects of cerebrospinal fluid (CSF) samples collected from animals treated with AMX/Clav (reference group) or with AMX/Clav/1,8-cineole (test group) against *E. coli* ESBL-producing strain growth (inhibition%). **: statistically significant difference compared to reference group (*p* < 0.01); ***: statistically significant difference compared to reference group (*p* < 0.001).

**Table 1 antibiotics-11-01294-t001:** Mean pharmacokinetic parameters obtained in the reference (AMX/Clav) and test groups (AMX/Clav/1,8-cineole) after the administration of treatments. **: statistically significant difference compared to reference group (*p* < 0.01).

Rabbit Groups	AUC_0–6h_ (µg.h/mL)	C_max_ (µg/mL)	T_max_ (h)	Ke	T_1/2_ (h)
Reference group	14.74 ± 0.9	3.49 ± 0.2	1.55 ± 0.1	0.32 ± 0.04	2.21 ± 0.3
Test group	22.30 ± 0.4 **	5.79 ± 0.2 **	1.18 ± 0.1	0.24 ± 0.01	2.94 ± 0.2

## Data Availability

The data used to support the findings of this study are available from the corresponding author upon request.

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
