# Peer review of "The Combination of Amoxicillin and 1,8-Cineole Improves the Bioavailability and the Therapeutic Effect of Amoxicillin in a Rabbit Model"

_antibiotics, 2022, doi:10.3390/antibiotics11101294_

Round 1
Reviewer 1 Report
The manuscript entitled "The Combination of Amoxicillin and 1.8-Cineole improves the 2 Bioavailability and the Therapeutic Effect of Amoxicillin in the 3 Rabbit Model" is a well-written research paper which suggests an interesting proposal for reducing the resistance to AMX/Clav. The parallel use of secondary metabolites by natural products may indeed be the right direction. However the purity of the metabolite isolated and the range of toxicity is very limited. In this paper the oral administration of AMX/Clav/1.8-cineole increments the inhibition % compared with reference group. Furthermore, it is very important to determine the toxic limit dose of 1,8-cineole in-vivo.
The metabolism of 1,8-cineole in mammals is well-known and there is a dependence by the incubation time and the its concentration. It is well known that monoterpenoids are metabolized by liver microsomes (especially oxidation and hydroxylation). Did the authors considered these parameters in the bioavailability of 1,8 cineole alone and in complex so to stabilize the maximum no toxic dose based on the weight?
Furthermore a more detailed information on the HPLC chromatograms is needed.
Author Response
The authors would like to thank all reviewers for their time and relevant suggestions and criticisms, which improved the quality of our manuscript.
- Comment 1: The manuscript entitled "The Combination of Amoxicillin and 1.8-Cineole improves the 2 Bioavailability and the Therapeutic Effect of Amoxicillin in the Rabbit Model" is a well-written research paper which suggests an interesting proposal for reducing the resistance to AMX/Clav. The parallel use of secondary metabolites by natural products may indeed be the right direction. However, the purity of the metabolite isolated and the range of toxicity is very limited. In this paper the oral administration of AMX/Clav/1.8-cineole increments the inhibition % compared with the reference group. Furthermore, it is very important to determine the toxic limit dose of 1,8-cineole in-vivo.
Response: According to the literature, the LD50 value for oral administration of 1,8-cineole was found to be 1280 mg/kg in the rat model (Jalilzadeh-Amin, & Maham, 2015). Oral administration of 1,8-cineole at 100 mg/kg did not cause any visible toxic symptoms, respiratory distress, ataxia, convulsion, or mortality in the animals (Jalilzadeh-Amin, & Maham, 2015). The LD50 values in other species have been reported to be 2300 mg/kg (intramuscularly) in Guinea pigs, 1500 mg/kg (subcutaneously) in dogs, and 50-3500 mg/kg in mice (McLean et al., 2007). (Please see page: 8, lines: 201 -210)
The concentration used in the present work (10mg/kg BW) was considerably lower than the LD50 used in the previous studies mentioned above.
- Comment 2: The metabolism of 1,8-cineole in mammals is well-known and there is a dependence by the incubation time and its concentration. It is well known that monoterpenoids are metabolized by liver microsomes (especially oxidation and hydroxylation). Did the authors considered these parameters in the bioavailability of 1,8 cineole alone and in complex so to stabilize the maximum no toxic dose based on the weight?
Response: We do agree with the reviewer on this point. However, the most important point in our study was the bioavailability of the AMX when administered in a mixture with 1,8-cineole, and we have shown that the bioavailability of the AMX was increased. Therefore, we did not consider these parameters mentioned by the reviewer in this work.
- Comment 3: Furthermore, a more detailed information on the HPLC chromatograms is needed.
Response: The plasma concentration of AMX (µg/ml) was calculated by the following formula:
AMX (µg/ml) = (RE X Conc T XFDE X Act (%)) / (RT X 100)
With:
RE: Average area ratio of AMX/cefadroxil in the test solution
RT: Average area ratio of AMX/cefadroxil in the control solution
Conc T: Concentration (µg/ml) of AMX in the control solution
FDE: Dilution factor of plasma in the test solution
Act %: Activity of AMX reference standard (%)
(Please see page: 12, lines: 348- 356)
Table: Results of AMX concentrations (µg/ml) are given in the following table:
|
Time (h) |
Reference Group |
Test Group |
||||
|
Average (µg/ml) |
SD |
CV |
Average (µg/ml) |
SD |
CV |
|
|
T= 0 |
0.00 |
0.00 |
0.00 |
0.00 |
0.00 |
0.00 |
|
T= 1h |
3.24 |
0.22 |
0.25 |
5.73 |
0.20 |
0.23 |
|
T= 2h |
3.52 |
0.16 |
0.19 |
4.89 |
0.42 |
0.48 |
|
T= 3h |
3.10 |
0.30 |
0.35 |
4.26 |
0.20 |
0.23 |
|
T=6h |
1.20 |
0.12 |
0.14 |
2.10 |
0.15 |
0.17 |
Reviewer 2 Report
The paper “The Combination of Amoxicillin and 1.8-Cineole improves the bioavailability and the Therapeutic Effect of Amoxicillin in the Rabbit Model” describes the increased therapeutic effect of combination therapy with 1,8-Cineole due to increased bioavailability and half-life. The paper is interesting and important due to its significance in treating difficult-to-treat infections. However, the manuscript require improvements at several places before it is fit for publication.
Major comments-
1. The authors should use the traditional antibacterial methods for determining antibacterial activity, such as growth curves over 24hr period, and test the antibacterial activity of plasma and CSF
2. The authors utilized E coli strains, but it will be important to show this effect in other pathogenic strains of bacteria that commonly cross BBB, especially when trying to make a case for activity from CSF fluid.
3. The in vivo relevance of this combination therapy is not provided in the current analysis. The authors should show the effectiveness of this combination therapy in rabbits (by survival or pathogen burden analysis)
Minor comments-
Line 38 – Please rephrase to clarify the opening statement
Figure legends: Please include more details in the figure legends, including the statistical test used
1.8-cineole – Please correct the name to 1,8-cineole or use alternative names for clarity
The flow of the manuscript can be improved
Author Response
The authors would like to thank all reviewers for their time and relevant suggestions and criticisms, which improved the quality of our manuscript.
Major comments
- Comment 1: The authors should use the traditional antibacterial methods for determining antibacterial activity, such as growth curves over 24hr period, and test the antibacterial activity of plasma and CSF
Response: We do agree with the reviewer that the traditional antibacterial methods are of great importance. However, due to the following technical limitations: repetitive samples, and the volumes of blood and CSF needed were not enough. Therefore, it was not possible to perform traditional techniques. Moreover, the animals remain alive throughout the experiment. Thus, we were obliged to use a microtechnique. (Please see page: 8, lines: 188-191)
- Comment 2: The authors utilized E coli strains, but it will be important to show this effect in other pathogenic strains of bacteria that commonly cross BBB, especially when trying to make a case for activity from CSF fluid.
Response: We do agree with the reviewer. However, the purpose of this assay was to examine the ability of the tested AMX, alone or in combination, to cross the blood-brain barrier (BBB). Therefore, we used the Escherichia coli ESBL strain as a model organism to evaluate the antibacterial activity of AMX in plasma and CSF. Furthermore, E. coli is one of the most common pathogens causing meningitis (Please see page: 2, lines 46-47)
- Comment 3: The in vivo relevance of this combination therapy is not provided in the current analysis. The authors should show the effectiveness of this combination therapy in rabbits (by survival or pathogen burden analysis)
Response: The present findings support previous studies published in our laboratory in which we have demonstrated a high synergistic effect between Amoxicillin/clavulanic acid and 1,8-cineole in vivo using an experimental model of methicillin-resistant S. aureus that causes osteomyelitis in rabbits (Hriouech et al., 2020). The effectiveness of the treatment was assessed by counting the number of bacteria in the bone marrow. A significant reduction in the number of bone marrow colonies was observed after four days of treatment when rabbits were administered AMX with 1,8-cineole compared to AMX alone (Hriouech et al., 2020). (Please see page: 8, lines: 214- 217)
https://doi.org/10.1155/2020/4271017
Minor comments
- Line 38 – Please rephrase to clarify the opening statement
Response: We have rephrased the opening statement. (Please see page: 2, lines: 38-39)
For many years now, the positive effects of antibiotics in the treatment of various infectious diseases were well documented
- Figure legends: Please include more details in the figure legends, including the statistical test used.
Response: We have added this sentence “The letters indicate a statistically significant difference (p < 0.05)’’ in the legends of figures 3 and 4. The details concerning the statistical test used are mentioned in the Materials and Methods section: the title "Statistical analysis" (Please see page:6, line:142 and page:7, line:159)
- 8-cineole – Please correct the name to 1,8-cineole or use alternative names for clarity.
Response: The name of 1,8-cineole was corrected. 1.8-cineole was replaced by 1,8-cineole in the manuscript
- The flow of the manuscript can be improved
Response: After responding and inserting the modifications requested by the five reviewers, the flow of the manuscript was improved.
Reviewer 3 Report
In the manuscript titled ‘The Combination of Amoxicillin and 1.8-Cineole improves the 2 Bioavailability and the Therapeutic Effect of Amoxicillin in the 3 Rabbit Model’ the authors performed a comparison of amoxicillin 16 (AMX) and clavulanic acid (Clav) with
AMX/Clav/1.8-cineole in rabbits. The plasma & CSF concentrations were significantly higher in AMX/Clav/1.8-cineole (test) than AMX and Clav (reference) group.
Major comments:
1) The authors have not demonstrated the mechanism of the increase of higher concentrations of test group drugs. Beta-lactams like penicillin is the substrate of OAT transporters and these transporters are present in the intestine, liver, kidney, and brain. The structures of penicillin and AMX are very similar. AMX is likely a substrate of OAT.
The authors should perform in vitro transporters assay to understand the mechanism. PMID: 25540139
2) Is 1.8-cineole an inhibitor of OAT transporter or inhibiting the metabolism of AMX?
3) Is there better solubility of AMX in the test formulation than reference formulation?
4) Did you quantify clavulanic acid and 1.8-cineole in plasma and CSF?
Author Response
The authors would like to thank all reviewers for their time and relevant suggestions and criticisms, which improved the quality of our manuscript.
Major comments
- Comment 1: The authors have not demonstrated the mechanism of the increase of higher concentrations of test group drugs. Beta-lactams like penicillin is the substrate of OAT transporters and these transporters are present in the intestine, liver, kidney, and brain. The structures of penicillin and AMX are very similar. AMX is likely a substrate of OAT.
The authors should perform in vitro transporters assay to understand the mechanism. PMID: 25540139
Response: We do agree with the reviewer, that studying the mechanism of action is of great importance. However, it is out of the scope of this work, which was to verify the bioavailability of AMX in the presence of the 1,8-cineole. The performance of an in vitro transporter assay to understand the mechanism warrants a separate in depth study.
- Comment 2: Is 1.8-cineole an inhibitor of OAT transporter or inhibiting the metabolism of AMX?
Response: In our previous in vitro study, we evaluated the effect of AMX/1,8-cineole association on the ESBL enzymatic resistance mechanism using a new optimized enzymatic assay. The results of this assay showed that the combination of AMX with 1,8-cineole notably influenced the enzymatic resistance reaction by decreasing the affinity of the AMX, to the ß-lactamase enzyme (Akhmouch at al., 2020). AMX recognition changes in the presence of 1,8-cineole according to these results. Therefore 1,8-cineole could decrease the affinity of OAT transporters. The mechanism remains to be elucidated. (Please see page: 9, lines: 257-263)
- Comment 3: Is there better solubility of AMX in the test formulation than reference formulation?
Response: No obvious changes were observed during the experiment in our conditions.
- Comment 4: Did you quantify clavulanic acid and 1.8-cineole in plasma and CSF?
Response: No. Clavulanic acid and 1.8-cineole were quantified neither in plasma nor in CSF, because our study was the focused effect on the bioavailability of the AMX.
Reviewer 4 Report
Akhmouch et.al, have described the use of combination of Amoxicillin and 1.8-Cineole to combat ESBL bacterial resistance.
Comments:
1. Is there a specific reason to choose rabbit as the animal model? If yes, please explain. Do you expect to have some species difference between rabbit and humans. If yes, please provide a comment in discussion section how to translate this data to humans. 2. How was the dose of 10mg/Kg of 1.8-cineole selected? Please provide rationale 3. Please provide a brief explanation of how is 1.8-cineole helping to improve absorption? What are the potential mechanisms? 4. Page 2, section 2.1 wrong figure number is referenced. 5. What gradient was used for HPLC? Please mention the details of the gradient 6. Please provide gender of the animals 7. What does superscript ‘a’ and ‘b’ mean in all the tables and figures? 8. What is AMO-4275 in figure 1C? 9. Line 107, add the word ‘times’ after half 10. What is the unbound fraction of 1.8-cineole in plasma and brain? 11. How were the different drugs dosed? Were they co-administered or dosed with some time gapAuthor Response
The authors would like to thank all reviewers for their time and relevant suggestions and criticisms, which improved the quality of our manuscript.
- Comment 1: Is there a specific reason to choose rabbit as the animal model? If yes, please explain. Do you expect to have some species difference between rabbit and humans? If yes, please provide a comment in the discussion section how to translate this data to humans.
Response: Rabbit was more adequate for the performance of the present experiments than mice and rats, because the size of the rabbit allows us to recover enough quantities of physiological fluids. Yes, we are expecting such results in humans, because the animal model is just to test the effect of 1,8-cineole in vivo using plasma or CFS to observe their activity on the E.coli strain. The clinical trials have been carried out and they will be published soon. (Please see page: 8, lines: 188-191)
- Comment 2: How was the dose of 10mg/Kg of 1.8-cineole selected? Please provide rationale
Response: The dose of 1.8-cineole (10mg/Kg) was selected based on the clinical trial performed by Kardos et al. (2021). He treated 132 patients diagnosed with ”acute bronchitis” with 3 × 200 mg of Cineole (Soledum®), which is the equivalent of 600 mg of 1,8-cineole for adults weighing 60 kg (equivalent to 10 mg/kg body weight). https://doi.org/10.1186/s40816-021-00319-8
- Comment 3: Please provide a brief explanation of how is 1.8-cineole helping to improve absorption? What are the potential mechanisms?
Response:
The two molecules 1,8-cineole and AMX are slightly hydrophobic and the fact that their association forms a complex more hydrophobic than either of the two molecules. It can cross the membrane more easily (Remmal, 2019). This does not prevent us from doing a deeper analysis, which will be the objective of a future studies.
- Comment 4: Page 2, section 2.1 wrong figure number is referenced.
Response: We checked that and the figure number was correct. (Please see page: 10, lines: 271- 274)
- Comment 5: What gradient was used for HPLC? Please mention the details of the gradient
Response: The mobile phase was a mixture of phosphate buffer (0.01 mol/L), pH = 4.8 and ACN (95: 5 v / v), pumped, in isocratic mode, at a flow rate of 1.3 mL/min through the column (Lichrosorb® 10µm RP 18, 250 x 4.6 mm; Phenomenex, France) at room temperature. (Please see page: 12 lines: 345-347)
- Comment 6: Please provide gender of the animals
Response: the gender of the animals is female. It was mentioned in the “animals” section. (Please see page: 11, line: 303)
- Comment 7: What does superscript ‘a’ and ‘b’ mean in all the tables and figures?
Response: Letters a, b, c, etc. were added to the Figures and Tables to show statistically significant differences between the variables. (Please see page: 6, line: 142 and page: 7, line: 159)
- Comment 8: What is AMO-4275 in figure 1C?
Response: AMO-4.275 is the retention time of amoxicillin in the chromatographic system.
- Comment 9: Line 107, add the word ‘times’ after half
Response: The word “times” after half was added. (Please see page: 4, line: 109)
- Comment 10: What is the unbound fraction of 1.8-cineole in plasma and brain?
Response: The unbound fraction of 1.8-cineole in plasma and brain was not studied. We were only interested in AMX.
- Comment 11: How were the different drugs dosed? Were they co-administered or dosed with some time gap
Response: The drugs (AMX/Clav and 1.8-cineole) were firstly mixed together and then administered to the animals. This was mentioned in the material and methods section. (Please see page: 10, lines: 287- 290)
Reviewer 5 Report
1. The internal standard (IS) peak in the HPLC chromatogram is after the drug peak and the baseline is not normal after the drug peak, in that case, the area calculated for the internal standard may not be correct. May have to rework the HPLC method.
2. Though partial validation was done what is the reason for performing partial validation? For a new HPLC method, a Pharmacokinetic study (PK) full validation should be performed.
3. What was the rationale for selecting the UV absorbance at 229 nm, because the λmax for the drug is 341nm and for IS is 249 nm.
4. For oral dosing it would be better to mention the formulation details.
5. After the blood collection to separate the plasma, in the collection tubes anti-coagulant should be added and it is not mentioned.
6. The time points collected are not sufficient to calculate t1/2, still the concentration was significant and later time points should also be collected to calculate t1/2the profile is not looking like oral dosing.
7. The linearity range was not mentioned and kindly mention the lower limit of detection too.
8. Please mention the reference for an anti-microbial activity for 1,8 Cineole.
Author Response
The authors would like to thank all reviewers for their time and relevant suggestions and criticisms, which improved the quality of our manuscript.
- Comment 1: The internal standard (IS) peak in the HPLC chromatogram is after the drug peak and the baseline is not normal after the drug peak, in that case, the area calculated for the internal standard may not be correct. May have to rework the HPLC method.
Response: No manual integration was used; data processing for all injections (Control solutions and Test solutions) was conducted by automatic processing method through EMPOWER 3 software.
The plasma concentration of AMX (µg/ml) was calculated by the following formula:
AMX (µg/ml) = (RE X Conc T XFDE X Act (%)) / (RT X 100)
With:
RE: Average area ratio of AMX/cefadroxil in the test solution
RT: Average area ratio of AMX/cefadroxil in the control solution
Conc T: Concentration (µg/ml) of AMX in the control solution
FDE: Dilution factor of plasma in the test solution
Act %: Activity of AMX reference standard (%)
(Please see page: 12, lines: 347- 355)
- Comment 2: Though partial validation was done what is the reason for performing partial validation? For a new HPLC method, a Pharmacokinetic study (PK) full validation should be performed.
Response: The referenced HPLC method described by Pires et al (2003) has been validated for the determination of plasma AMX concentrations in human plasma. However, our study consists in using rabbits as an animal model instead of the human species while keeping the same biological matrix (plasma). Therefore, a prior validation of this method was performed according to the bioanalytical method validation guidelines (FDA, 2018). (Please see page: 11, lines: 330-331)
- Comment 3: What was the rationale for selecting the UV absorbance at 229 nm, because the λmax for the drug is 341nm and for IS is 249 nm.
Response: The rationale for selecting the UV absorbance was based on the use of Pires’s HPLC method (Pires et al 2003) which describes the UV detection at 229nm.
- Comment 4: For oral dosing it would be better to mention the formulation details.
Response:
1,8-cineole, in liquid form, was dispersed in a viscous solution of 0.2% (v/v) agar according to the method described by Remmal et al (1993). The AMX/Clav was dissolved in sterile distilled water and stirred until totally dispersed. Then the two solutions were mixed. This was described in the material and methods section (Please see page: 10, lines: 283- 292)
- Comment 5: After the blood collection to separate the plasma, in the collection tubes anti-coagulant should be added and it is not mentioned.
Response: Indeed, the blood samples were taken in tubes already containing the anti-coagulant lithium heparin (2.5 ml, AX3430, LABO-MODERNE France) (Please see page: 11, lines: 319-320)
Comment 6: The time points collected are not sufficient to calculate t1/2, still the concentration was significant and later time points should also be collected to calculate t1/2 the profile is not looking like oral dosing.
Response: The half-life elimination of Amoxicillin after oral administration has been reported by many studies. In animals, AMX has a mean elimination half-life of approximately 1 hour (Carceles et coll, 1995; Del et coll, 1998; Yang F et coll, 2019). (Please see page: 9, lines: 248-250)
- Comment 7: The linearity range was not mentioned and kindly mention the lower limit of detection too.
Response: The calibration curve was linear over the range 1.0 µg/ml to 50µg/ml, with a regression coefficient ≥ 0.999 and intercept not significantly different from zero. The LOD and the LOQ for AMO were 0.1 and 1µg/ml, respectively.
- Comment 8: Please mention the reference for an anti-microbial activity for 1,8 Cineole
Response: we have added the reference for the anti-microbial activity of 1,8 Cineole in the manuscript. (Please see page: 8, lines: 192-201, and page: 8, lines: 211-217)
Previous research data revealed that 1,8-cineole was a strong active compound against many pathogenic strains, including Escherichia coli, Staphylococcus aureus, Pseudomonas aeruginosa, and Bacillus subtilis (Papadopoulos et al., 2008; Merghni et al., 2018). Moreover,1,8-Cineole had a high effect against Bacillus cereus and Cryptococcus neoformans at a dose of 2 mg/ml (Magiatis et al., 2002). A trial conducted by Moo et al. (2021), illustrated that 1,8-cineole has potent activity on carbapenemase-producing Klebsiella pneumoniae (KPC-KP) at the concentration of 28.83 mg/mL. In this study, the authors demonstrated that 1,8-cineole induced oxidative stress and membrane damage resulting in KPC-KP cell death. This was observed by scanning and transmission electron microscopies that confirmed bacterial cell membrane rupture and loss of intracellular materials (Moo et al., 2021).
Over the past years, researchers have shown great interest in the association of 1,8-cineole with chemical antimicrobial agents such as chlorhexidine digluconate (Hendry et al., 2009) and mupirocin (Kifer et al., 2016). This aimed to enhance their antibacterial activities, in the treatment of vulvovaginal candidiasis, bacterial vaginosis, and acne (Trinh et al., 2011). Findings in vitro by Kwiatkowski et al. (2020) and those in vitro and in vivo by Hriouech et al. (2020) revealed the synergistic effect of 1,8 cineole in combination with beta-lactam antibiotics including penicillin G and AMX respectively against MRSA strains.
Round 2
Reviewer 2 Report
The authors have adequately responded to my comments, and I recommend its publication.
Author Response
The authors would like to thank the reviewer for his time and relevant suggestions and criticisms, which improved the quality of our manuscript.
Reviewer 3 Report
Lines 72 and 73 check the units, they don't match the figure 2
Author Response
- Comment 1
Lines 72 and 73 check the units, they don't match the figure 2
Response: The authors would like to thank the reviewer for his consideration and attention. We have corrected the units. please see lines 72 and 73.
‘’The LOD and the LOQ for AMX were 0.1 and 1 μg/mL, respectively. The calibration curve was linear over the range of 1.0 μg/mL to 50 μg/mL’’
Reviewer 4 Report
1. Figure 1C, use AMX 4.275 instead of AMO-4,275 to be consistent with the rest of the paper
2. Use asterisk instead of letters to indicate significance. Using letters is confusing
3. It will be useful to the readers to know if 10mg/kg dose reached plasma concentrations above the minimum effective concentration. If yes, please provide a statement regarding that along with a reference of minimum effective concentration. The addition of discussion regarding LD50 is useful to know the above limit however it will be also useful to know what is lower limit (what is the no effect level and was it considered while deciding the dose)
Author Response
- Comment 1
Figure 1C, use AMX 4.275 instead of AMO-4,275 to be consistent with the rest of the paper
Response: The modification was done. (Please see page 3).
- Comment 2
Use asterisk instead of letters to indicate significance. Using letters is confusing
Response: The authors would like to thank the reviewer for his careful attention. We have replaced the letters with asterisks in figures and tables to indicate significance. (Please see lines 97-98 on page 4, lines 115-116 on page 5, lines 9134-135 on page 6, and lines153-154 page 7).
*: statistically significant difference compared to reference group(p<0.05)
**: statistically significant difference compared to reference group (p<0.01)
***: statistically significant difference compared to reference group (p<0.001)
- Comment 3
- It will be useful to the readers to know if the 10mg/kg dose reached plasma concentrations above the minimum effective concentration. If yes, please provide a statement regarding that along with a reference of minimum effective concentration. The addition of discussion regarding LD50 is useful to know the above limit however it will be also useful to know what is lower limit (what is the no effect level and was it considered while deciding the dose)
Response:
The total blood volume of the rabbit weighing 2kg is approximately 100 ml and the plasma represents about 50% (Greenfield, 2022). We administered 20 mg of 1.8-cineole/rabbit and if this concentration could totally reach the plasma it means that 20mg of cineol will be found in 50 ml of plasma. This corresponds to 0.4 mg/ml. Thus, this dose is below the MIC of 1.8-cineole which was 28.83mg/mL approved by Moo et al., (2021). In our study we showed the antibacterial effect of plasma against E. coli, this means that the dose of 1,8-cineole reached the plasma was sufficient to boost AMX and enhance its effect.
Moreover, to obtain the synergy between 1,8-cineole and AMX, only a dose of 1,8-cineole varied between 6.2 and 7.1 mg/L were needed (Akhmouch et al., 2022).
Reviewer 5 Report
The study is a continuation and supporting of the previous work and in future it is better to quantify in LC-MS/MS.
Author Response
- Comment 1
The study is a continuation and supporting of the previous work and in the future, it is better to quantify in LC-MS/MS.
Response: The authors would like to thank the reviewer for all of his comments and suggested edits. They are much appreciated.